# Beyond the Decline of Wild Bees: Optimizing Conservation Measures and Bringing Together the Actors

**DOI:** 10.3390/insects11090649

**Published:** 2020-09-22

**Authors:** Maxime Drossart, Maxence Gérard

**Affiliations:** Laboratory of Zoology, Research Institute for Biosciences, University of Mons (UMONS), Place du Parc 20, B-7000 Mons, Belgium

**Keywords:** wild bees, insect pollinator, decline, conservation actions, conservation tools

## Abstract

**Simple Summary:**

Wild bees represent the main group of pollinators in Europe, being responsible for the reproduction of numerous flowering plants. However, like a non-negligible part of biodiversity, this group has been facing a global decline mostly induced by numerous human factors over the last decades. Overall, even if all the questions are not solved concerning the causes of their decline, we are beyond the precautionary principle because the decline factors are roughly known, identified and at least partially quantified. Experts are now calling for effective actions to promote wild bee diversity and the enhancement of environmental quality. In this review, we present a general and up-to-date assessment of the conservation methods, as well as their efficiency and the current projects that try to fill the gaps and optimize the conservation measures. This publication aims to be a needed catalyst to implement concrete and qualitative conservation actions for wild bees.

**Abstract:**

Wild bees are facing a global decline mostly induced by numerous human factors for the last decades. In parallel, public interest for their conservation increased considerably, namely through numerous scientific studies relayed in the media. In spite of this broad interest, a lack of knowledge and understanding of the subject is blatant and reveals a gap between awareness and understanding. While their decline is extensively studied, information on conservation measures is often scattered in the literature. We are now beyond the precautionary principle and experts are calling for effective actions to promote wild bee diversity and the enhancement of environment quality. In this review, we draw a general and up-to-date assessment of the conservation methods, as well as their efficiency and the current projects that try to fill the gaps and optimize the conservation measures. Targeting bees, we focused our attention on (i) the protection and restoration of wild bee habitats, (ii) the conservation measures in anthropogenic habitats, (iii) the implementation of human made tools, (iv) how to deal with invasive alien species, and finally (v) how to communicate efficiently and accurately. This review can be considered as a needed catalyst to implement concrete and qualitative conversation actions for bees.

## 1. Introduction

Over the last decade, public interest in the conservation of pollinators increased considerably, thanks in particular to numerous scientific studies relayed in the media [1,2,3]. While some less-charismatic insect taxa remain unknown from public appreciation [4], marketing pollinator conservation is an easier sell to a broad audience. However, while this interest seems to be already acquired for a majority of the population, a lack of knowledge and understanding of the subject is blatant and thus reveals a gap between awareness and understanding [2]. This gap is particularly characterized by a general lack of awareness of the great diversity of pollinators (e.g., bees, hoverflies, butterflies…). For example, respectively 80% and 99% of survey respondents in Great Britain and the United States claim that bees are important, but only 3% and 14% can assess the number of species found in their country [2,5]. Moreover, most citizens are able to identify honeybees and bumblebees as bee species, but the other wild bee species are poorly recognized as bees [2]. Less than 50% of the respondents are even unable to name at least one bee species [5]. Through massive communication, but mainly focused on the honeybee *Apis mellifera*, the general audience understood the importance of pollination, but not the phenomenon and its magnitude as a whole (e.g., the importance of combining floral resources and nesting sites, the existence of different major groups of pollinators with sometimes specific requirements, etc.). Such a lack of knowledge could therefore lead, on the basis of good intentions, to irrelevant actions, misguided, or even counterproductive measures for the protection of threatened pollinator species [2]. The misinterpretation of scientific articles, sometimes amplified by media, can even accentuate these inappropriate actions. Clearly, the awareness of public audience has not grown with the increase of scientific research on this topic. Yet for scientists, identifying and prioritizing threatened populations is challenging and still subject to ongoing investigation [6]. As a basis for all researchers, Harvey and colleagues, [7] formulated an informative road map for insect conservation and recovery.

Overall, the abundance of terrestrial insects has indeed deceased by 9% per decade and the wild bees are not an exception to the rule [8]. Literature about the deleterious impact of stressors on wild bees is abundant [9,10]. To slow down their negative impacts, the development of national and regional initiatives has been more and more associated with the implementation of restoration and protection actions for pollinators for the last few years. All of these conservation actions undertaken locally are of paramount importance in the preservation of local biodiversity. Indeed, while many species are threatened on a local scale and even suffer regional extinction, they may still remain quite common within their overall geographical range [3,11,12]. This phenomenon can be observed for instance by comparing global (e.g., European scale) and regional IUCN Red Lists [13]. As an example, many bee species from Belgium are (nearly) threatened or even extinct at a regional scale while their populations are not preoccupying at European scale (i.e., LC, Least Concern) [14,15] (Figure 1). This paradox highlights the importance of conservation measures at various geographical scales.

While these species status as well as the decline factors are extensively studied, information on wild bee conservation measures is often scattered in the literature. A general and up-to-date assessment of the conservation methods, as well as their efficiency, is still lacking. Based on this, we first briefly reviewed the importance of conserving wild bees, the assessments of the bee populations at risk as well as the associated decline factors. Then, we focused our attention on the conservation measures, the associated actors, and the efficiency of these actions. More specifically, we highlighted the conservation measures that increase the quality of bee-friendly habitats among a wide range of landscapes, from semi-natural habitats to urban and agricultural landscapes. We underlined the needed floral and nesting resources, as well as possible habitat managements. We then tackled the case of alien species and ended up by the ways to communicate and educate the public. Throughout this review, we pointed out the recent projects and studies that tried to fill the numerous gaps of wild bee conservation. These notions are articulated around a virtuous circle of conservation scheme represented in Figure 2.

## 2. Why Should We Conserve Wild Bees?

Pollinators are involved in the fecundation of approximately 80% of flowering plants, bees being the primary pollinators in most ecosystems [16,17,18]. This is why the pollination service can be considered as one of the most important services in the context of ecosystem functioning but also in global agricultural production [18,19]. Indeed, most agricultural crops (roughly 85%) benefit from animal pollination that directly affects the yields and/or production quality [18,19,20]. This pollination service to food production has been evaluated to be worth between 100 to 500 billion euros each year worldwide [20,21,22,23]. The loss of bee pollinators is thus expected to induce a shift in human food consumption to non-pollinator dependent crops and lead to deficiencies in crucial nutrients, with clear economic and health issues [19,24,25,26]. Wild bees can be the major pollinators in terms of abundance in some crops [18], while other crops are predominated by managed honeybees or other insect taxa [27,28]. However, only a small part of wild bee species forage on crops. Thus, the maintenance of pollination services alone is not enough to justify alone wild bee conservation [29]. Wild bees are also of particular importance due to their ability to forage in a wide range of weather conditions and habitats [30]. In particular, bumblebees are generalist and can forage at low temperatures [31] but are also able to forage on flowers requiring high frequency sonication [32]. Seemingly, in wild plant communities, the high diversity of wild pollinators and the high degree of specialization among some species strengthen the complementarity and the synergism with honeybees [33,34,35]. In this respect, Grab et al. [36] highlighted that the phylogenetic diversity of bees is the explanatory criteria related to the quality of the pollination service in agricultural areas, before abundance and species diversity. In parallel, functional diversity also plays a key role to promote crop production [37]. Finally, in other ecosystems like urban areas [38], forests [39], or natural ecosystems [39,40], the crucial role of wild bees has also been demonstrated and their diversity is often tightly linked to the diversity of wild plants [41,42].

As namely highlighted by Valiente-Banuet et al. [43], the extinction of ecological interactions between species, such as pollinators and plants, are often neglected instead of the assessment of species extinction. Yet, this component of biodiversity loss often occurs in parallel or may even precede species extinctions [43,44]. Concerning bees, some species are closely linked to particular habitat and/or food resources (i.e., specialist), making them the prompt to rapidly suffer from habitat/food disruptions [45]. Indeed, specialist bees forage on a narrow range (or unique) of plant taxa whereas generalist bees tend to be nonspecific in their foraging choices [46,47,48]. The latter’s group appear then to be less vulnerable to fluctuating environment conditions like those linked to human activities as they are able to forage on alternative food resources [49,50].

The loss of these biotic interactions could then have pervasive effects accelerating species extinctions and negative impact on ecosystem functions through their importance in key functional aspects of ecosystems [51]. In order to prevent a collapse of the provided ecosystem services to humans, the component of biotic interactions should then be considered (e.g., [44]) to assess the ‘health’ of ecosystems and to detect potential environmental issues [52,53]. Considering this, several projects, like the EU FP7 BeeFun project in Andalucía, are aiming to improve knowledge and understanding of the impact of climatic events on plant-pollinator networks and communities [1].

## 3. Preliminary Steps: Assessment of the Extinction Risk and Knowledge Enhancement

The set-up of conservation measures requires a preliminary step for which data can be scattered: the assessment of population and species trends. Indeed, global change can either harm or benefit species. Depending on the context, species can be winners or losers of this global change, as recently suggested by Powney and colleagues, [54] in the United Kingdom. Based on a set of criteria, IUCN Red List methodology allows to evaluate a risk of extinction of a species at a global, national, or regional level [13]. Depending on the level of assessment, the methodologies can somewhat vary. However, it is the most relevant tool to build a substantial basis for further conservation measures and evaluate the trends of the species to later implement monitoring, conservation planning, management and decision making [55]. In wild bees, Nieto et al. [14] produced a Red List at European level and more and more countries are producing their own Red List at national and regional level e.g., [15,56,57]. Other IUCN assessments are conducted for specific groups (e.g., *Bombus* spp.) at continental scale, in North America for example [58]. This growing information for wild bees helps to highlight the populations, species, and areas that are more at risk.

Setting up conservation plans and strategies also demands to document factors threatening the populations [7,12,18]. The scientific community has focused on assessing the extent of the decline and studying the factors responsible for the population regressions [59]. Aside from natural hazards recently shown to have a non-negligible impact on wild pollinators (e.g., fire, drought, hydrological and geophysical events; [60]), global warming, agricultural intensification, habitat homogenization as well as diseases and pathogens strongly affect wild bee populations [9,18]. While each of the factors alone [9] as well as some of the possible interactions [10,61] have been widely studied in several continents worldwide, a strong knowledge gap lies probably in the quantification of the respective impacts from the different threats coupled with the use of historical collections [62]. The spatial and temporal correlations of most of the decline factors render the assessment particularly challenging.

Moreover, the understanding of evolutionary processes such as the genetic variation, population dynamics and speciation are also crucial and increasingly used in conservation biology [63,64]. Advances in conservation genetics allowed scientists to use molecular tools to improve their understanding of bee diversity [65,66], with new genomic methods also emerging [67]. This is particularly true for cryptic species or species with high level of morphological convergence such as bumblebees. Lower genetic diversity is also an early warning that can lead to inbreeding depression and decrease the fitness [68,69]. Based on phylogeographic studies and isolation by distance analyses (IBD), the assessment of the connectivity between populations, their effective population size as well as the biotic and abiotic drivers of these parameters, scientists can prioritize the populations at risk [70,71,72,73]. Based on population assessments through different tools and the evaluation of the threatening factors, experts are now calling for conservation actions through policy consensus, applied strategies, and initiatives in parallel with ongoing research [8,59].

## 4. Conservation Measures and Actors

Once population trends and decline drivers have been identified, the next step is to preserve favorable habitats. Indeed, there is a high heterogeneity of the wild bee specific richness and abundance depending on the scale and the landscape [74,75]. Worldwide, wild bee composition is structured by landscape composition from mountainous tropical ecosystems of Colombia [76] to dry grasslands in Missouri, USA [77]. Flowering grasslands present, for instance, a higher bee richness and abundance than agricultural areas dedicated to livestock and dominated by forage crops [78]. Thus, the first tool to preserve of these favorable habitats can be achieved by the establishment of protected areas via legal measures to prevent additional alterations, as well as the purchase of ecological valuable areas. In these protected areas, adaptive management of both pollinators and ecosystems have to be set up by testing assumptions to define the best strategies and adapting the measures consequently to empirical evidence of the successes and failures of the management measures [79,80,81]. However, many anthropogenic habitats (e.g., cities) cannot have the same amount of protection than semi-natural areas, but yet require conservation measures. In order to reach these goals, a range of strategies targeting the protection and restoration of ecosystems, the enhancement of biodiversity, the decrease of invasive alien species or the communication about diversity loss have been adopted from global (e.g., The World Bee Project which aims to combine cloud computing with bee research across the world to provide to all new insights and knowledge in order to find solutions to the global bee and biodiversity decline, climate change and increase food security and livelihoods) to continental scale with, among others, strategies undertaken by the European Union (i.e., EU Biodiversity Strategy to 2020, [82]) addressed to be then applied and adapted at national scale e.g., [83] as well as at different levels to counter or mitigate the decline of pollinators (e.g., in USA, [84]; in France, [85]; in Ireland, [86]). Non-governmental (e.g., Seeds for Bees, Bee and Butterfly Habitat Fund) as well as Governmental programs (e.g., The Dutch Bee Strategy, the English National Pollinator Strategy) have help to gather stakeholders from various backgrounds to share their expertise and act jointly [87,88]. To perpetuate these initiatives, the implication of public audience, notably through citizen sciences and the participation of youth is essential [88]. In the following sections, we detailed the management measures that can be conducted from protected areas, to more anthropogenic landscapes. We centered this discussion on (i) the protection and the restoration of wild bee habitat, (ii) a focus on anthropogenic habitats like urban and agricultural areas, (iii) the implementation of human-made tools to provide nesting resources, (iv) the potential issues of alien species and finally, (v) the communication as well as the education of the public audience.

### 4.1. Protection and Restoration of Wild Bee Habitats

First, wild bee conservation can be conducted by the protection of (semi-) natural environments in order to (re)constitute large connected habitats dedicated to biodiversity. Habitat protection is crucial: in a recent meta-analysis, it has indeed been shown that the decline of terrestrial insects was weaker in protected areas [8]. Methods to define theses protected areas are diverse: the most common try to take into account the greater specific diversity (e.g., [89]). More complex methods assume that this metric alone is too static and should encompass changes in climate and species distribution. These dynamic components can notably be assessed using Ecological Nich Models as Krechemer and Marchioro [90] did for several species of South-American bumblebees. In these protected areas, ecological restoration of habitats can increase wild bee abundance and richness across a wide range of landscapes and geographical [91]. The enhancement of the overall habitat quality by restoration measures implies an investigation of the habitat and resources preferences of the targeted wild bee species. For example, Svensson et al. [92] assessed the preferences of bumblebees in agricultural landscape and concluded that field boundaries and open uncultivated landscapes favored their populations. Depending on their ecological traits, wild bee abundance can indeed respond differently in different habitat types. Actually, Carrié et al. [93] showed that grasslands, hedgerows, and forest edges positively selected traits like solitary and ground-nesting bees visiting a wide range of flower species, while the abundance of social species as well as species nesting above the ground and visiting a narrow range of flower species decreased in these habitats. Conversely, the increase of rainforest cover could increase above-ground nesting species as described in Brazil [94]. It illustrates the importance to identify what we want to conserve to apply the corresponding restoration measures, depending on the type of landscape. Moreover, the restoration methods can be double-edged and are often context-dependent. For instance, while grazing and burning are often used in grasslands because they can help floral blooming [29], they can also disturb wild bee populations by killing overwintering individuals in the vegetation [91]. Such restoration actions are notably undertaken in the framework of LIFE projects financed by the European Union. They aim to restore environments and habitats of species included in the “Birds” and “Fauna-Flora-Habitats” directives in Natura 2000 sites. If insect conservation remains at its early stages compared to its big brothers, namely bird and mammal conservation, some LIFE projects (e.g., LIFE Butterflies) target pollinating species in particular such as butterflies and others (e.g., LIFE in Quarries) will indirectly benefit to wild bees with the protection and the restoration of qualitative habitats (e.g., peat bogs, quarries, meadows of high biological value, hedgerows, etc.) [95]. Even if LIFE dedicating to wild bees have long been absent, projects like Urbanbees LIFE project (i.e., LIFE + biodiversity program) are recently emerging (www.urbanbees.eu) and are aiming at the development of a management guide to maintain and increase the abundance and diversity of bees in urban and peri-urban environments. However, this LIFE project does not focus on natural/seminatural environment and it is the exception rather than the rule. The absence of bees in conservation programs is also perceptible at the political scale. For instance, Hall and Steiner [96] outlined that US state policies mostly ignore insect pollinators compared to vertebrates in consequence to a weak understanding of their requirements, leading to inadequate measures. Few conservation plans focus on pollinators in semi-natural habitats such as reserves and are taken into account in programs at national/sub-national scale; this first step should be implemented in further conservation programs and specifically identify their targeted species.

### 4.2. Conservation Measures in Anthropogenic Habitats: The Examples of Urban and Agricultural Areas

Conservation actions are crucial in urban areas since more than 55% of the population lives in (peri-) urban areas [97]. The transition towards a monocultural system is typically known to reduce wild bee abundance and diversity [98]. Measures are thus particularly important in highly urbanized or intensively managed habitat because these landscapes often lack food resources and nesting sites. By combining ecological, economic (i.e., cost) and logistical (i.e., implementation, sustainability of the management process) interests, their management can be adapted and thus enable the valorization of these neglected areas. Promoting protected areas close to sites with high anthropogenic pressures could therefore act as a buffer for wild bee populations [99]. Consequently, some initiatives are emerging such as guides stimulating the creation and management of favorable habitats to pollinators in agricultural environments through applied advices [100,101,102,103]. Roadsides, hedgerows, parks, and urban gardens can represent important habitats for wild pollinators, both qualitatively and quantitatively, as well as transition zones for habitat connectivity [104,105,106,107]. These areas can indeed support a high specific diversity and an increasing proportion of rare species [108]. They represent a transition between different types of habitats, and often support large bee diversity [109,110]. In Amsterdam, a bee-friendly strategy, promoting roof-top gardens, parks, and roadsides, has shown its capacity to increase wild bee diversity [111]. Moreover, this habitat connectivity is crucial, notably because habitat fragmentation could be particularly detrimental for smaller bees. Indeed, body size is often correlated to dispersal abilities [112]. In fragmented habitats, smaller bees can be unable to reach patches of suitable habitats and thus particularly suffer in anthropogenic landscapes [113,114]. These elements allowing connectivity represent relatively preserved habitats providing nesting sites and floral resources in the scope of urbanization or intensive agriculture [115,116]. However, while there is a theoretical framework about the importance of the connectivity elements, it has not been empirically tested. It is crucial if we want to know the efficiency of these measures.

Aside from urban areas, conservation strategies in agricultural areas like crops and farmland have still demonstrated a positive impact, depending on the type of measures [117], the targeted taxa [118] and the landscape composition [119,120]. Conservation actions in this type of habitat is crucial since almost 40% of land use is devoted to agriculture worldwide [121]. Restoration of floral resources is one of the most common measure to conserve wild bee anthropogenic habitats. It has widely been demonstrated that richness of floral resources is a key parameter for wild bee diversity, notably when restoring prairies like in Minnesota, USA [122]. However, several measures can only affect a narrow range of the biodiversity and occur at local scale [123]. For example, agri-environmental actions like flower strips have been initiated in Europe to promote biodiversity in intensively managed agricultural landscapes [124,125]. Yet, these seem to benefit mainly generalist species such as bumblebees, honeybees, and hoverflies (in Germany, [126]; in Belgium, [127]; in England, [128]). The increase in flower availability often results in an increase of bumblebee population size (i.e., an increase in colony density) and of the number of offspring [129,130]. The results based on Agri-Environmental Schemes (AES) came to the same conclusions. Introduced in the 1980s, the impact of AES was rarely tested until recently [131]. Among them, Geppert et al. [132] tested the impact of organic farming and flower strips. Overall, both measures were positively correlated with pollinators’ richness and abundance as well as bumblebee colony growth, but the effectiveness of these measures depended also strongly on the surrounding landscape [132]. Wood et al. [133] also evaluated a specific agri-environment scheme (i.e., Higher Level Stewardship farms—HLS) to test if sown flowers benefited to wild solitary bees in England. They found that only one third of the bees collected in the studied area effectively forage on sown flowers from agri-environment scheme seed mixes, thus advising to re-think the composition of seed mixes if we want to maintain diverse wild bee communities. Depending on the type of habitat and the plant species reintroduced, wild bee taxa will indeed show different responses. For example, while honeybees and bumblebees were positively affected by the occurrence of *Phacelia* sp., solitary bees mostly visited sunflower and wildflower seed mixtures [134]. A study conducted in Great Britain confirmed that seed mixtures sowed in agricultural sector contain a high proportion of clover species *Trifolium* sp. (Fabaceae), which is highly attractive to bumblebees, but not for most of the 240 other solitary bee species in that country [135]. It contrasts with the importance of diverse pollinator communities needed to optimize the ecosystem service of pollination [34,36,136,137]. The choice of plant mixtures thus depends on the conservation target. Such agri-environmental measures beneficial to a reduced number of pollinating species have been developed to primarily meet the economic interests of the agricultural sector (i.e., to ensure a secure income for producers) and are essentially based on the needs of generalist species that are widely used in crop pollination [126,138,139,140,141].

Moreover, it seems that the research carried out in this framework e.g., [142] has served as a basis and reference for the conservation of all pollinators among many initiatives (e.g., recommendations in the framework of some National Pollinator Strategy [143]). Yet, a shift in floral resources occurred during the last decades is among the most common stressor affecting wild bee habitats and could lead to floral resources depletion as well as changes in plant-pollinator networks [50,144]. The choice of plants in floral mixes has thus to be completely redesigned, because this choice is often lead by studies about honeybee and bumblebee visitation frequency [88,145]. The redesign of floral mixes could also be particularly efficient in habitats with strong floral resources turn over through space and time (e.g., agricultural areas and heathlands [146,147]). However, conservation practitioners still lack of information about the empirically tested measures; their choices are thus often driven by the measures benefiting from important commercial advertisement [88]. The choice of floral species should for example take into account the abundance of floral resources throughout the day as well as during the whole flying season [148,149] depending on the species-specific behavior and phenology. Gresty et al. [150] showed that species like *Rosa canina*, *Malva sylvestris*, and *Ranuncula acris* particularly attracted cavity-nesting wild solitary bees. Thus, most of the wildflower seed mixtures do not match the requirements of cavity-nesting wild solitary bees as they do not include these species. The choice of plant species should also consider their nutrient content. This parameter is crucial for bee populations but is often ignored when choosing the bee-friendly plants. It is particularly true for the larval nutritional requirements, which are different from adults [149]. Higher plant species richness as well as the occurrence of plant species with high nutritional values (e.g., *Brassica napus* for *Osmia bicornis*) can in turn positively influence population growth rate [151,152,153]. Indeed, the diversity of proteins and essential amino acids needed is crucial for colony growth and development. To avoid these nutritional deficiencies, the ecological stoichiometry can be evaluated by the atomic rations of crucial elements (e.g., C, K, P, N and Na) for the development of larvae [154]. These ratios are poorly known and remain to be assessed in a wide range of plant taxa. Some initiatives such as the biodivERsA NUTRIB2 research project (2020–2023) are emerging and focus on this important topic. It could help us to ultimately support the health of wild bee species by providing and/or protecting nutritionally suitable floral resources in environments, especially in resources-depleted ones. Some additional agricultural methods could also be more deeply investigated, e.g., companion planting, which can increase the yield production and production quality (e.g., the strawberry *Fragaria x ananassa* and the borage *Borago officinalis*, [155]) while the impact on pollinator diversity was not tested.

Finally, concerning pesticide use, most of the knowledge consists of their impacts on honeybees as well as several species of bumblebees and information about their sub-lethal and lethal effects on wild bees are scarce. Synergistic effects can also occur and are even less studied [10,156]. Because of their larger body size, bumblebees consume a lower dose of pesticide per gram than honeybees [157]. However, they also visit two or three times more flowers, are active during a wider range of climatic conditions, their larvae are fed with raw unprocessed pollen and nectar while consuming far more pollen than honeybee larvae [75]. It is thus hard to use the conclusion about pesticides toxicity drawn from the studies on honeybees and generalize them on bumblebees as well as wild solitary bees. Moreover, the sensitivity of solitary bees seems to be highly variable [158]. European projects like POSHBEE could help us to better understand how pesticides can impact these wild species as well as the synergism with other decline factors. For now, in most cases, the precaution principle is recommended and alternative for pesticides through new (or restored) management practices can help to this transition [159]. For example, the use of some plant essential oils [160,161] or the development of new biomolecules aiming at combating crop diseases with respect to ecosystems (see the Interreg SMARTBIOCONTROL portfolio of projects, http://www.smartbiocontrol.eu/) could represent valuable alternatives. We also have to test for reduced application timing and chemicals from biological origins [162]. Integrated Pest Management (IPM) is promoted as a way to mitigate the negative impact of intensive agricultural management on bees. Yet, it has been shown to also impact pollinators and is not a specific response to protect bee populations [163,164]. Egan and colleagues [164] thus introduced a new systematic framework that is called Integrated Pest and Pollinators Management (IPPM; [165]), in order to integrate measures specifically benefiting pollinators. They propose different measures to (1) avoid reaching action thresholds for both pest and pollinators but also (2) curative measures once the action thresholds have been exceeded. For instance, they propose practical measures like the selection of cultivars with high pollinator attractiveness and high pest resistance. Some other initiatives have been recently proposed to develop sustainable strategies in agricultural landscapes at various geographical levels. Indeed, agricultural habitats are probably the type of landscape where most of the current projects try to provide solutions. For example, the European EcoStack H2020 project aims at developing and supporting sustainable crop production and protection strategies (PPS) by considering the three ecological, economic and social main aspects to enhance sustainability in the systems of food production in Europe. At a national scale, the Protecting Farmland Pollinators project in Ireland is based on the development of a whole-farm pollinator scoring system which can allow farmers to easily identify what management practices on farmlands benefit to pollinators (e.g., reduction to suppression of pesticides input, provide small habitats and food resources such as flowering layouts at the farmland scale). This project is then about the enhancement of the farming system quality by the implementation of actions allowing to biodiversity to co-exist. In taking actions to protect these important species, farmers start a virtuous chain reaction for environment, and the wellbeing of future generations. In addition, the Interreg-Sudoe Poll-Ole-GI project (2016–2019) aimed at identifying and recommending effective methods such as green infrastructures (GI) to positively impact on pollinator communities and pollination ecosystem service in the two most important Mediterranean crops of arable farmland in the Southwestern European Space (SUDOE) which includes Southern France, Spain, and Portugal: sunflower and oilseed rape (https://pollolegi.eu/; [1]).

### 4.3. Providing Nesting Resources

In recent years, more attention has been paid to improve the availability of floral resources for pollinators across scientific literature, whereas few studies have focused on the other essential aspect: nesting sites [166]. In practice, little effort has concentrated about the (a)biotic factors influencing nesting success or site selection by different species [167,168]. Moreover, the true positive or negative impact of the developments carried out for bees is still poorly quantified. A higher proportion of threatened, endangered or already extinct species could be found among soil-nesting bees or bees with particular nesting behavior, such as species nesting in existing cavities below ground, carder bees and species nesting in snail shells e.g., [15]. However, the overwhelming majority of the nesting arrangements generally recommended and carried out simply consist in the installation of insect hotels benefiting only a small proportion of common and widespread species. Indeed, the study of Fortel et al. [166] highlights the use of insect hotels in urban areas by 21 species, including 17 species of Megachilidae. This represents less than 10% of the species richness observed in the same area (i.e., 248 species observed in 16 urban and peri-urban sites) [169]. In parallel, Maclvor and Packer, [170] also assessed the potential of this new habitat augmentation schemes. They highlighted that nearly half of the bees nested in the hotels were introduced/exotic bees, but also that 75% of insects occupied it were wasps. These results are in line with the study of Geslin and colleagues, [171] highlighting the high occupation rate of bee hotels by the exotic *Megachile sculpturalis* (i.e., 40% of all individuals in 96 bee hotels). More worryingly, they also observed a negative correlation between the occurrence of this species in bee hotels and the presence of native bees. Aside from these issues, bee hotels could also facilitate the transmission of diseases [170,172]. These studies thus highlight that the considerable effort to promote this conservation tool is maybe not as beneficial as previously thought [171]. Considering hypogean nesting, Fortel and colleagues [166] tested 6 types of soil texture in soil squares to evaluate their use in the same context. While texture did not seem to influence species richness, 37 species were observed in these squares, most of them being Andrenidae and Halictidae species. In total, nearly 20% of the bees observed in the area (i.e., 57 species out of 248) used the man-made structures [166]. This study demonstrates the positive potential of man-made implements for wild bees, but also highlights the importance of diversifying nesting resources e.g., [168,173,174] to promote diverse bee communities in the management of green spaces. Indeed, attention could namely be paid to the diameter of the holes used by species nesting in existing cavities (e.g., bee hotels): small diameters could favor many native bee species and avoid larger exotic bees like *M. sculturalis* to nest in the bee hotel [171]. Whether it is the installation of small patches of bare soil or the installation of bee hotels, empirical evidence of their positive impact remains to be deeply explored.

### 4.4. Dealing with Invasive Alien Species

In addition to the native plant species that can be implemented by practitioners through conservation measures, several invasive alien species can colonize bee-friendly habitats, or even be voluntarily planted. These invasive alien plants can have contrasting effects on wild bees: positive, neutral, or even negative. Surely, the impact of invasive alien plant species can depend of several factors like ecological context or life history traits [81], leading some species to suffer from the invasion [50] while others can have some benefit from it [50,175,176]. However, an invasion only occurs when an exotic plant with invasive potential encounters a sufficiently favorable environment [177]. Indeed, the sensitivity of ecosystems to invasions is variable and is influenced in particular by (i) the degree of disturbance (i.e., openings created in ecosystems and structuring vegetation communities) and (ii) the resource availability (i.e., particularly in mineral elements that influence competition processes) [177,178,179]. This is namely the case when vegetation is removed following a physical disturbance or when an exogenous supply of nutrients induces eutrophication [177,179]. The latter appears to represent the predominant factor related to invasion phenomena [179,180,181] and would thus have led to an overall impoverishment of the flora [182,183]. In general, most of these plant species have then logically become established in habitats closely associated with human activity [184]. For example, more than 64% of naturalized exotic plants in Europe are found in industrial areas, 58.5% in farmland and parks and gardens, 37.5% in lawns, and 31.5% in woods and forests [184]. In the context of highly forage-depleted environments associated with human activities, invasive plant species can provide, in some cases, valuable food resources for generalist pollinators and restore/enhance ecological functions such as pollination [69,81,185,186,187]. For instance, in Ohio, USA, *Trifolium repens* and *Trifolium pratense* help to restore wild bee populations in urban areas ([188,189].

However, they can also have detrimental effects on specialist bee species with low diet plasticity [81,190]. Indeed, some invasive alien plants compete e.g., [191,192] with native flora and can replace plant species that are foraged by specialist pollinators e.g., [193,194]. Some non-native species also display resources that are not accessible to most of wild bee species (e.g., *Petunias* sp., [195]). Conservation measures to combat alien plant species could thus be applied accordingly to the environmental context and the precautionary principle. They may range from the eradication of the invasive population to the control of this population. In a recent meta-analysis, Majewska and Altize [196] finally showed that no systematic positive or negative impact on pollinator abundance could be attributed to exotic species and that their impact is case-specific. The postulate that all exotic species are harmful is thus as detrimental to the knowledge enhancement as is the axiom that these species are harmless [197].

Considering invasive pollinators, their occurrence can lead to direct (e.g., the competition for food and nesting resources or the transmission of pathogens [198,199]) or indirect (the modification of food web and plant communities’ structure [81]) effects. One of the main examples is the Asian hornet *Vespa velutina* which could represent a potential threat for wild and/or domesticated pollinators in the coming years [200,201,202]. Escaped alien bumblebees used for agricultural pollination is also a growing concern. Indeed, the consequences of the growth of the bumblebee trade are increasingly pointed out as a major issue for global diversity [203]. The introduction of *Bombus terrestris* in non-native geographical range can also result in mating with native species and potentially the production of non-viable offspring, as it has been observed in Japan with *B. hypocrita* and *B. ignitus* [204]. In south America, the introduction of *Bombus terrestris* and *B. ruderatus* has notably induced the decline of a native species, *B. dahlbomii*, as well as pathogen transmission [205]. While several countries developed measures as banning of bumblebee trade of alien species, these policies have not always been uniformized. Coordinated international measures to avoid biological invasions should thus be a central concern in importation policies. The setup of a monitoring of these invasive species represents a priority: it could allow to obtain the data needed to follow population dynamics and assess their potential impact on native species and ecosystems [206].

At the European level, a regulation to avoid and mitigate the adverse effects of invasive alien species in the European Union came into effect in 2015 (EU Regulation (EU) No. 1143/2014). It defines a set of preventive and curative measures to apply to any organism included in the list which actually counts 66 species. The implementation of these measures aims at preventing, or at least reducing and mitigating harmful effects of these species through a comprehensive, coordinated, and effective response to the problem of biological invasions in Europe. It is based on close collaboration between all Member States and requires enhanced cooperation with economic sectors, non-governmental organizations and citizens. As with the IUCN Red Lists assessment, a similar classification system called the “Black List” has been proposed for invasive species and considers the degree of environmental impact for the species assessed [207]. It includes species with moderate, major, and severe environmental impact. Finally, the IUCN Invasive Species Specialist Group (ISSG) gathers experts in the aim of reducing threats caused by invasive alien species to native ecosystems and the species they contain by increasing awareness, prevention, control, or eradication concerning them. Adapted at a national scale, each group is responsible for preparing and updating the reference list of invasive species in their own country.

### 4.5. Conserving Wild Bees: An Action for All

From the accumulating knowledge and the efficiency of conservation measures, communicate more efficiently and accurately scientific results to broader audience is a keystone to establish solid conservation basis. Saunders et al. [208] stated clearly how we should choose our words to bridge the gap between awareness and understanding. They highlighted the importance to use clear terminology and unambiguous concepts in communication, by namely specifying the taxonomic and geographic scale as well as the aim and the results of a research program.

In term of communication and education, some priorities can be pointed out to build support for insect conservation [95,209]. First, it is crucial to develop citizen sciences programs combining education and training with the collect of long-term data. In different taxa, extensive use of citizen sciences in conservation programs allowed monitoring populations over long time series [210,211]. During the last decade, a growing public interest has focused on pollinators [15]. In spite of significant biases in records (i.e., recording of the most striking and colorful species, the misidentification of cryptic species) obviously occurring with this method, it allows to efficiently map a huge number of specimens at larger scale, representing a meaningful part of the species in an area and helping to evaluate the species trends in the scope of scientific works e.g., [212] and Red Lists e.g., [15,213,214]. The development of citizen sciences can also be favored by the development of new technologies allowing insect identification. Then, an increasing knowledge of the public is directly linked to the improvement of school, education and training programs to increase the value of natural history observation and the knowledge for all (e.g., schoolchild, students, green space managers). Indeed, Silva and Minor, [215] highlighted that the level of education and knowledge about bees were directly related to the positive behaviors of the respondents towards the bees, emphasizing the crucial role of public awareness. Finally, the creation of media ethics training for scientists in order to deliver clear communications focused on scientific methods and processes is essential. The last point constitutes a critical part of the research process and impacts directly on the engagement of the public in the nature conservation. This communication to the population has to be broad (e.g., publishing results in scholarly literature, social media, etc.) but also accurate in the facts (e.g., reporting the geographic and taxonomic scale of the results) [209]. One of the most common misunderstanding of the society is that bees are primarily associated with the words “honey”, “hives”, and “sting”. This is one of the challenges in the communication about bees: directly link the notion of bees with pollinators and wild species than with the aforementioned notions [216]. Turo and Gardiner [88] also highlighted that one of the main challenges to achieve wild bee conservation in urban areas is that urban residents need to feel safe and find the measures aesthetic if we want to hope for long term measures. The support of residents is crucial to avoid vandalism and collect funding. In green public areas, there is indeed a general preference for clear edges, mown grass, and a global neatness of the area. To address this resident preference, the setting-up of areas dedicated specifically to agroecosystems and the establishment of “pocket prairies” is advised [88].

Practically, in addition to the biodiversity monitoring provided by these programs, citizens and other stakeholders can inform themselves to take effective conservation actions [95]. In (sub)urban habitats, listing recommended plants promoting both generalist and specialist bees depending on their nutritional value as well as the local context can support citizens but also local or municipal authorities in their choices. Technical knowledge (e.g., favoring alternative concepts instead of the use of agrochemicals, mowing/pruning calendar) represents also a major step to effectively integrate pollinators in the management scheme of all green spaces (e.g., parks, gardens, flower stripes) [95]. Austria represents a good example with several projects aiming at positively impact on wild pollinators with namely the creation/restoration of around 20ha of flowering areas by 18 local authorities accompanied by a monitoring of wild bees [1]. Beside this, pilot projects are leaded to create suitable habitats in schools, nurseries, orchards, road edges, and along waterways [1]. As the conservation of bees, and biodiversity in general is everyone’s business, actions can also be implemented by hand in hand and namely by public-private partnerships. As example, a partnership was set up between an NGO and a supermarket chain in Austria, between fruit farmers and public municipalities in Flanders (Belgium) or between a beer brewer, NGO, and public authorities [1].

## 5. Conclusions

In conservation biology, the urgency of species conservation requires experts to express opinions and advice based on the evidence available at some point [12,217]. Based on the assessment of extinction risks of species (i.e., through IUCN red list and citizen sciences monitoring), targeted species requiring conservation measures can be pointed out. Through this review, we highlighted that the first step is to clearly define these targeted species. Indeed, they will display different ecological traits, floral and nesting specific requirements that need to be identified if they have to be preserved. The effect of conservation measures also has to be evaluated not only in terms of specific diversity and abundance, but also in terms of these ecological and functional traits; these aspects are largely ignored while functional diversity is known to be crucial in ecosystems (e.g., [218]). Many conservation measures are often conducted based on empirical evidence on very particular species, namely honeybees and bumblebees, but conclusions based on these studies can rarely be used as proxy for wild solitary bees [219]. The choice of floral species in seed mixes and flower strips clearly illustrates this problematic. While has been known for a long time that floral resources are a central topic of wild bee conservation, the majority of the papers highlighted that the plant species occurring in the mixes are only foraged by a small proportion of bee species. For instance, Nichols et al. [135] highlighted that a total of 14 species of flowering plants belonging to 9 families (e.g., Asteraceae, Apicaeae, Geraniaceae) would attract 37 of the 40 species of bees encountered in agricultural environments and would collect nearly 100% of the floral visits. However, only two of these species are already included in the sowed mixtures recommended in national strategies for the conservation of pollinators e.g., [143]. We think that a first central point is to completely redesign these mixes taking into account new empirical studies identifying foraged plant species (e.g., [150]), matching the flying season of targeted bee species and the nutritional requirements of both larvae and adults [149]. For now, this point is only vaguely investigated when designing conservation programs. Moreover, there is a serious disequilibrium between the studies about floral resources and those about nesting resources. As both points have to be addressed in conservation programs, further studies need to investigate this point more deeply, notably in testing a wider range of nesting resources than only bee hotels that host a small part of bee species. In a world that is increasingly anthropogenically-driven, alternative floral resources could be, punctually, a solution. In environments with very few bee-friendly native plants, some exotic invasive species can help populations to recover. Again, these choices should be taken based on strong empirical evidence since a recent study in New-York suggested that many exotic plants could mainly help honeybees more than wild bees [220]. In these anthropogenic environments, the way to manage bee-friendly areas have also to be re-think, notably by finding alternatives to pesticides. In this context, the impact of methods like IPPM could be deeply investigated. In addition, the overwhelming majority of habitat restoration studies are conducted in North America and Europe, which represent only a small part of terrestrial habitats inhabiting by bees. These measures could not be representative of what should be done elsewhere on different type of landscapes [91]. Further studies need to test conservation measures in a wider range of habitats like rainforest and arctic regions. Finally, in the context of these accumulating empirical evidence, the communication and the education about bees have to be reshaped. Everybody knows that you do not conserve birds by building henhouses, we thus have to redouble our efforts so that the lexical field of bees does not revolve around the words beehives and honey. To do this, we should not forget to include work packages in our research projects about the transmission of information to schools and those who are practically working to implement conservation measures. These work packages are often the ones that are sacrificed while they should be the culmination of many scientific projects.

Despite accumulating knowledge about threatening factors will always be valuable, we are beyond the precautionary principle [221] because these are mostly known, identified, and at least partially quantified [59,95]. Experts are now calling for actions, which represents the major priority [7,18,59,95]. Recent research projects like the Interreg SAPOLL project (i.e., cross-border project with the development of an action plan for the conservation of pollinators as the main goal, www.sapoll.eu) or the biodivERsA NUTRIB2 project could help to reduce these gaps. The optimization of these actions and bringing together the different conservation actors would reinforce the public awareness to biodiversity and ecosystem services in environments where people are increasingly disconnected from nature, such as in urban areas [166,222].

## Figures and Tables

**Figure 1 insects-11-00649-f001:**
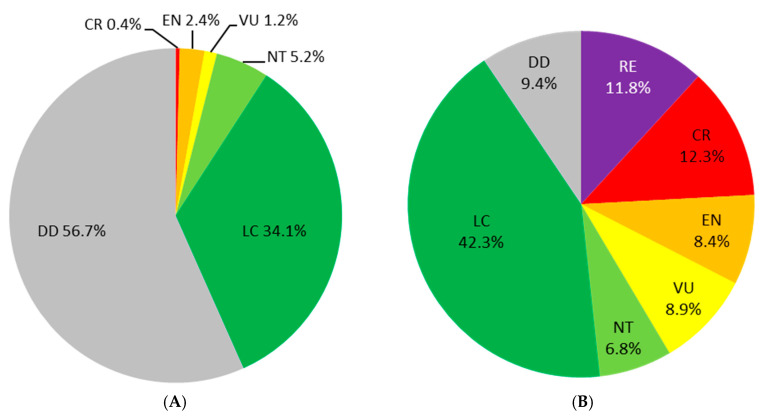
IUCN Red List status in (**A**) Europe [14] and (**B**) Belgium [15]. The proportion of (nearly) threatened and data deficient species is different and highlights the importance to implement conservation actions at various scales. LC = Least Concern, NT = Near Threatened, VU = Vulnerable, EN = Endangered, CR = Critically Endangered, RE = Regionally Extinct, DD = Data Deficient.

**Figure 2 insects-11-00649-f002:**
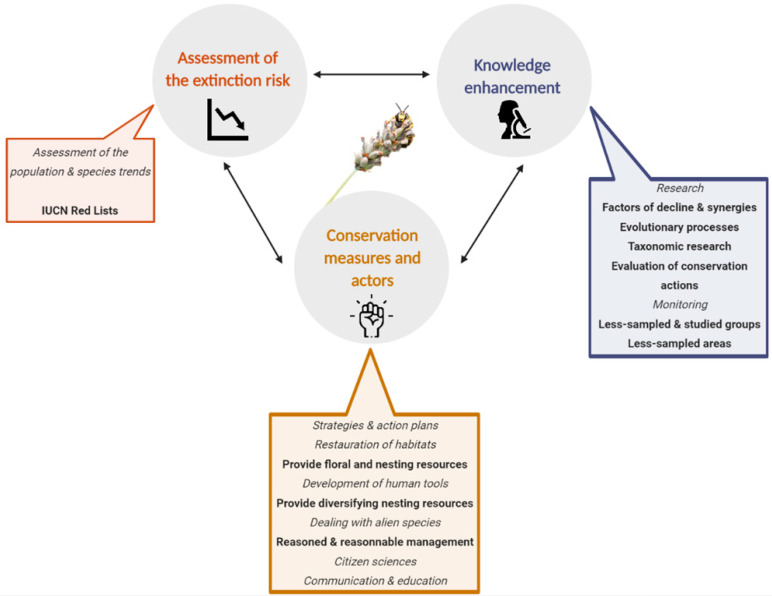
Virtuous circle of a conservation scheme implying to undertake simultaneously these distinct three steps. Icon made by Freepik perfect from www.flaticon.com.

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
