# Peer review of "Beyond the Decline of Wild Bees: Optimizing Conservation Measures and Bringing Together the Actors"

_insects, 2020, doi:10.3390/insects11090649_

Round 1

Reviewer 1 Report

The Authors did a very good job at correcting their manuscript and thanks to their efforts it is now a sound and relevant review of our knowledge and knowledge gaps. There is only a minor question/correction left I would like to ask/suggest. 

line 57 - 60 The Authors use the example of European and regional red lists. I cannot agree, specifically in this case, that bees are less threatened on a EU scale than on a regional scale. The truth is, that almost 60% of the bee species present in EU has a data deficient status (DD) - we simply have no idea how many are out there or if they actually still exist. One cannot actually compare these two scales, because the EU scale is simply too data deficient to say anything about "EU bees". We can only say, that our knowledge on a regional scale is much more detailed. Unfortunately, not all EU countries have such good bee monitoring and data as Belgium (only 9.4% DD). The EU bees with DD status can all be extinct or threatened, we simply do not know what is going on with them. The EU Red list Data actually shows a huge gap in our knowledge. Although, I agree, that this data does show the importance of conservation measures on various levels.

Author Response

Reviewer #1 (Comments and Suggestions for Authors):

The Authors did a very good job at correcting their manuscript and thanks to their efforts it is now a sound and relevant review of our knowledge and knowledge gaps.

-- Many thanks for this comment

There is only a minor question/correction left I would like to ask/suggest. 

line 57 - 60 The Authors use the example of European and regional red lists. I cannot agree, specifically in this case, that bees are less threatened on a EU scale than on a regional scale. The truth is, that almost 60% of the bee species present in EU has a data deficient status (DD) - we simply have no idea how many are out there or if they actually still exist. One cannot actually compare these two scales, because the EU scale is simply too data deficient to say anything about "EU bees". We can only say, that our knowledge on a regional scale is much more detailed. Unfortunately, not all EU countries have such good bee monitoring and data as Belgium (only 9.4% DD). The EU bees with DD status can all be extinct or threatened, we simply do not know what is going on with them. The EU Red list Data actually shows a huge gap in our knowledge. Although, I agree, that this data does show the importance of conservation measures on various levels.

-- We totally agree with the fact that there are contrasts between scales (EU – national) and between countries. This is namely largely discussed in the Belgium Red List of Bees (Drossart et al. 2019) and a table compares the proportion of assessed and threatened species between countries and, as you mention it, there is a huge difference between EU countries. However, among the assessed species at both continental and national scale (we don’t take into account the 56.7% of DD species), some are not threatened by considering their whole range (EU scale) instead of the administrative limits of countries. We highlighted the fact that for one species (e.g. Anthophora aestivalis assessed as RE in Belgium but LC in Europe), the assessment can be different according to the scale. And as we finish the paragraph, this paradox highlights the importance of conservation measures at various scales.

We clarified it in the text by adding this above mentioned example and replacing “many bee species from Belgium …” by “some bee species from Belgium”.

Reviewer 2 Report

I would like to thank the authors for their work. In my opinion this manuscript may be published after considering the below comments:

Line 23: it should be ‘conservation’ instead of ‘conversation’.

Line 83: please make a new paragraph after the fig.2 legend.

Lines 173-182: this sentence is very long and therefore hard to understand. Please make a few simpler sentences instead

Lines 196-198: I don’t understand. What do you mean? Please clarify.

Lines 201-202: please clarify the following part of this sentence: ‘across a wide range of landscapes and geographical’.

Line 219: please remove ‘big brothers’; please be specific in your writing.

Line 315: please remove ‘(e.g. Brassica napus for Osmia bicornis)’ since B. napus pollen has low nutritional value for O. bicornis.

Lines 364-369: this sentence is too long. Please make 2-3 shorter sentences instead.

Line 380: please remove ‘the study of’ and please avoid fillers in your writing.

Lines 525-576: I guess this whole paragraph was formatted as a numbered list by mistake. If so, please reformat.

Lines 529-531: this sentence is a bit strange in my opinion. Are you sure it should end with ‘pointing out’ ?

Author Response

Reviewer #2 (Comments and Suggestions for Authors):

I would like to thank the authors for their work. In my opinion this manuscript may be published after considering the below comments:

Line 23: it should be ‘conservation’ instead of ‘conversation’.

-- It has been edited.

Line 83: please make a new paragraph after the fig.2 legend.

-- It has been edited.

Lines 173-182: this sentence is very long and therefore hard to understand. Please make a few simpler sentences instead

-- It has been edited.

Lines 196-198: I don’t understand. What do you mean? Please clarify.

-- It has been clarified.

Lines 201-202: please clarify the following part of this sentence: ‘across a wide range of landscapes and geographical’.

-- It has been clarified.

Line 219: please remove ‘big brothers’; please be specific in your writing.

-- It has been removed and edited.

Line 315: please remove ‘(e.g. Brassica napus for Osmia bicornis)’ since B. napus pollen has low nutritional value for O. bicornis.

-- It has been removed

Lines 364-369: this sentence is too long. Please make 2-3 shorter sentences instead.

-- It has been edited.

Line 380: please remove ‘the study of’ and please avoid fillers in your writing.

-- It has been removed and edited.

Lines 525-576: I guess this whole paragraph was formatted as a numbered list by mistake. If so, please reformat.

-- It has been reformatted.

Lines 529-531: this sentence is a bit strange in my opinion. Are you sure it should end with ‘pointing out’ ?

-- It has been edited.

This manuscript is a resubmission of an earlier submission. The following is a list of the peer review reports and author responses from that submission.

Round 1

Reviewer 1 Report

A fine and well timed review on an interesting topic.

I have only a few remarks:

Figure 1: I suggest to make clear all acronyms as you did at line 54 for Least Concern

Line 691: checki the justification

Line 693: remove the I before Insect Conserv. Diver.

the manuscript is a review paper which discusses a much departed topic: how to prevent wild bee decline? Of course all the claims in the paper are not novel, since they are scattered through the scientific literature, but they are collated in a convincing way. The authors did an extensive literature search and the outstanding references are properly discussed.
The authors focus on wild bees, while many similar papers mostly discuss honey bees.
Honey bees are key pollinators in their native range and many feral colonies survive in the wild, but most honey bee colonies are managed and honey bees were introduced by man in the Americas, East Asia and Australia were they somehow compete with native bees. Also some other bee species are managed and moved by man outside their native range, but to a far lesser extent than honey bees. All these facts are clearly stated and briefly discussed in the paper.
The authors provide a fair overview of the state of the art and put the topic into the general field of nature conservation. Such an approach is rather unusual, but it is the correct one. To a certain extent the authors focus on the European Union approach to the protection of wild bees, which they surely know better than others approaches, but the European Union approach is a good example of what should be done to protect wild bees.
Further work would certainly improve the paper, but not so much to be significant

Author Response

The manuscript is a review paper which discusses a much departed topic: how to prevent wild bee decline? Of course all the claims in the paper are not novel, since they are scattered through the scientific literature, but they are collated in a convincing way. The authors did an extensive literature search and the outstanding references are properly discussed.
The authors focus on wild bees, while many similar papers mostly discuss honey bees.
Honey bees are key pollinators in their native range and many feral colonies survive in the wild, but most honey bee colonies are managed and honey bees were introduced by man in the Americas, East Asia and Australia were they somehow compete with native bees. Also some other bee species are managed and moved by man outside their native range, but to a far lesser extent than honey bees. All these facts are clearly stated and briefly discussed in the paper. The authors provide a fair overview of the state of the art and put the topic into the general field of nature conservation. Such an approach is rather unusual, but it is the correct one. To a certain extent the authors focus on the European Union approach to the protection of wild bees, which they surely know better than others approaches, but the European Union approach is a good example of what should be done to protect wild bees. Further work would certainly improve the paper, but not so much to be significant

-- Many thanks for this comment

A fine and well timed review on an interesting topic, I have only a few remarks:

Figure 1: I suggest to make clear all acronyms as you did at line 54 for Least Concern

-- We added all the acronyms in the figure legend lines 65-66.

Line 691: check the justification

-- The justification has been checked.

Line 693: remove the I before Insect Conserv. Diver.

-- It has been edited.

Reviewer 2 Report

Review paper “Wild bee conservation: a review of initiatives, measures efficiency and knowledge gaps” written by Maxime Drossart, and Maxence Gérard.

The Authors have undertaken the difficult task of writing a review paper on the wild bee conservation. A lot has been already said regarding this topic and it is hard to provide novel and needed information to the Reader. Unfortunately, the Authors have taken the easy way and provided too much information that was already discussed in recent reviews, including in the review published in Insects by Belski and Neelendra (2019). The Authors should familiarize with this important paper to prevent replication of information already provided there. Also, in the introductory part of the current review the Authors should briefly refer to recently published reviews and should clearly state what new and needed information they want to provide in the current review, why it is important and interesting to the
Reader. For example the current review might focus on emerging issues and the newest literature. In the current version of the manuscript the Authors focused on well-established knowledge only scanning through some new research in chronicle record manner (i.e. not really discussing the
meaning, importance and significance of recent advances) as well as describing marginally a few conservation initiatives, limited to the European Union, instead of providing real review of such initiatives. In my opinion, the current version of the manuscript is not suitable for publication,
however it represents a good start - a really good draft on which the valuable paper may be build, after major rewriting of the text and elaborating some specific issues (as suggested below). Therefore, I
suggest either resubmission of the improved version of the paper or major review, meaning deep and thorough rewriting of the paper, as suggested below. Since the manuscript, in my opinion, should be thoroughly rewritten, I provide only major comments.

I strongly encourage the Authors to familiarize with this short but helpful article before rewriting the manuscript: Sayer, E. J. (2018). The anatomy of an excellent review paper. Functional Ecology, 32(10), 2278– 2281. https://doi.org/10.1111/1365-2435.13207

General major comments:

Three main weak points of this paper, making it not a review but rather sort of opinion paper are:

(1) very narrow focus on European Union (covering approx. half of Europe’s area and this half is very specific regarding landscapes, habitats, agriculture, law, politics, summing up to specific nature conservation issues) and French-speaking countries inside the Union, which makes the paper of local
importance. The Authors should broaden their point of view in the improved version of the paper;

(2) the paper is very short as for review and neither of the topics covered by the Authors is sufficiently discussed (the majority of them is only mentioned without any discussion!). The Authors should focus on the most interesting and novel issues and discuss them deeply;

(3) in the current form the manuscript is not a coherent and gripping story. It is rather a collection of stories where one story does not led to another. The Authors should provide the clear line of though leading to the clear, concrete and precise conclusions saying what we can learn based on previous paragraphs and what new important knowledge is revealed when discussing previous sections in holistic and synergistic approach. Content wise, to focus on new knowledge that this manuscript could provide, I have paid particular attention to broadening its substantive effect. First, I provide most important recent literature that seems to be ignored by the Authors. I recommend to familiarize with the below mentioned studies and
reports:

Belsky, & Joshi. (2019). Impact of Biotic and Abiotic Stressors on Managed and Feral Bees. Insects, 10(8), 233. doi: 10.3390/insects10080233 – in-depth review regarding issues related to the bee conservation. It covers topics discussed in the current review, including a chapter on initiatives to
support bees;

Ollerton, J. (2017). Pollinator Diversity: Distribution, Ecological Function, and Conservation. Annual Review of Ecology, Evolution, and Systematics, 48(1), 353–376. doi:10.1146/annurev-ecolsys110316-022919 – this paper was cited by the Authors but it seems that the Authors do not read it
thoroughly. I strongly recommend detailed, close reading of this publication. The eco-evo context is necessary when discussing bee conservation (in the current text the Authors provide exclusively the economic/agricultural point of view and bees mean much more than this).

van Klink, R., Bowler, D. E., Gongalsky, K. B., Swengel, A. B., Gentile, A., & Chase, J. M. (2020). Meta-analysis reveals declines in terrestrial but increases in freshwater insect abundances. Science, 420(April), in press. doi: 10.1126/science.aax9931 – this paper should be considered when writing about insect decline.

And here are some papers (and one report) considering ignored aspects of bee conservation (filling some gaps) and initiatives to support bees:
Reversing the Decline of Insects, ed: Dave Goulson, The Wildlife Trusts 2020: https://bit.ly/3iKuQdJ more: https://www.wildlifetrusts.org/take-action-insects Turo, K. J., & Gardiner, M. M. (2019). From potential to practical: conserving bees in urban public green spaces. Frontiers in Ecology and the Environment, 17(3), 167–175. doi: 10.1002/fee.2015

Wood, T. J., Holland, J. M., & Goulson, D. (2017). Providing foraging resources for solitary bees on farmland: current schemes for pollinators benefit a limited suite of species. Journal of Applied Ecology, 54(1), 323–333. doi:10.1111/1365-2664.12718

Egan, P. A., Dicks, L. V., Hokkanen, H. M. T., & Stenberg, J. A. (2020). Delivering Integrated Pest and Pollinator Management (IPPM). Trends in Plant Science, 25(6), 577–589. doi: 10.1016/j.tplants.2020.01.006

Gresty, C. E. A., Clare, E., Devey, D. S., Cowan, R. S., Csiba, L., Malakasi, P., … Willis, K. J. (2018). Flower preferences and pollen transport networks for cavity-nesting solitary bees: Implications for the design of agri-environment schemes. Ecology and Evolution, 8(15), 7574–7587. doi:10.1002/ece3.4234

Filipiak, M. (2018). A Better Understanding of Bee Nutritional Ecology Is Needed to Optimize Conservation Strategies for Wild Bees—The Application of Ecological Stoichiometry. Insects, 9(3), 85. doi:10.3390/insects9030085

Filipiak, M. (2019). Key pollen host plants provide balanced diets for wild bee larvae: A lesson for planting flower strips and hedgerows. Journal of Applied Ecology, 56(6), 1410–1418. doi: 10.1111/1365-2664.13383

Carvell, C., Bourke, A. F. G., Dreier, S., Freeman, S. N., Hulmes, S., Jordan, W. C., … Heard, M. S. (2017). Bumblebee family lineage survival is enhanced in high-quality landscapes. Nature, 543(7646), 547–549. doi:10.1038/nature21709

Hall, D. M., & Steiner, R. (2019). Insect pollinator conservation policy innovations at subnational levels: Lessons for lawmakers. Environmental Science & Policy, 93(October 2018), 118–128. doi:10.1016/j.envsci.2018.12.026

Majewska, A. A., & Altizer, S. (2020). Planting gardens to support insect pollinators. Conservation Biology, 34(1), 15–25. doi: 10.1111/cobi.13271 Schönfelder, M. L., & Bogner, F. X. (2017). Individual perception of bees: Between perceived danger and willingness to protect. PLOS ONE, 12(6), e0180168. doi:10.1371/journal.pone.0180168

Wilson, J. S., Forister, M. L., & Carril, O. M. (2017). Interest exceeds understanding in public support of bee conservation. Frontiers in Ecology and the Environment, 15(8), 460–466. doi:10.1002/fee.1531

Silva, A., & Minor, E. S. (2017). Adolescents’ Experience and Knowledge of, and Attitudes toward, Bees: Implications and Recommendations for Conservation. Anthrozoös, 30(1), 19–32. doi:10.1080/08927936.2017.1270587

Colla, S. R., & MacIvor, J. S. (2017). Questioning public perception, conservation policy, and recovery actions for honeybees in North America. Conservation Biology, 31(5), 1202–1204. doi:10.1111/cobi.12839

Tonietto, R. K., & Larkin, D. J. (2018). Habitat restoration benefits wild bees: A meta-analysis. Journal of Applied Ecology, 55(2), 582–590. doi:10.1111/1365-2664.13012

Lane, I. G., Herron‐Sweet, C. R., Portman, Z. M., & Cariveau, D. P. (2020). Floral resource diversity drives bee community diversity in prairie restorations along an agricultural landscape gradient. Journal
of Applied Ecology, (January), 1365-2664.13694. doi: 10.1111/1365-2664.13694

Geppert, C., Hass, A., Földesi, R., Donkó, B., Akter, A., Tscharntke, T., & Batáry, P. (2020). Agrienvironment schemes enhance pollinator richness and abundance but bumblebee reproduction depends on field size. Journal of Applied Ecology, 1–11. doi: 10.1111/1365-2664.13682

Knowledge gaps are mentioned in the title. It may be interesting to discuss these gaps within the context of the ongoing research projects, which will provide new knowledge in the nearest future. The Authors could compare the projects and provide information on which knowledge gaps will be
probably filled in the nearest future and on which gaps we should focus now when writing new research proposals. For example, I know that following projects are ongoing within the EU (it’s the
Authors job to find more):
https://www.poshbee.eu/
https://nutrib2.project.uj.edu.pl/start
https://www.ecostack-h2020.eu/
https://ec.europa.eu/food/animals/live_animals/bees/research_en
and more, not strictly related to EU programs + worldwide:
http://worldbeeproject.org/
https://www.biodiversityireland.ie/
https://pollinators.ie/record-pollinators/bees/
https://ento.psu.edu/pollinators/stakeholder-advisory-board-on-pollinator-health

The Authors mention National Pollinator Strategy in the text. Since the manuscript is a review, it is not enough. The Authors should discuss various strategies related to pollinator conservation in different countries, compare them within the context of landscapes, habitats land use and agriculture
characteristics, that differ between countries and ultimately the Authors should provide advice on the realities of pollinator conservation adoption.

Detailed major comments:

Title is not informative: practically it means “wild bee conservation: a review”. I suggest to focus in the title on one strong take home message and make this message the central point of the manuscript.

Introduction – the Authors write about “lack of awareness of the great diversity of pollinators” which may lead to “irrelevant actions”, misguided measures, etc. And the rest of the manuscript concerns only bees. Why? Why the Authors first write about the great diversity of pollinators and then ignore it and write only about bees?

Chapters 1 and 2 are disconnected from the rest of the text. The manuscript could start from the chapter 3. Figure 1 – description of abbreviations is needed. The chapter 2: “Why should we conserve wild bees?” should be either removed (it is not necessary) or elaborated much more and completely rewritten. The Authors described only the commercial/agricultural importance of wild bees, which is very anthropocentric. Bees are much more than this – they play ecological roles in their environments and interact with different components of
food webs, etc.

The topic of wild bee conservation initiatives is central part of the title but was neglected in the text of the manuscript: the LIFE EU framework was mentioned in lines 135-149 and all the information provided in lines 150-205 are mainly hand-waiving, i.e. superficial, round-about and without any
specifics. This should be improved. Lines 170-195 and 333-337: the problem of appropriate floral composition of flower strips, beefriendly seed mixes etc. is very important. It seems that the majority of seed/flower mixes used are of not appropriate composition for wild bees, focusing on nectar-rich plants for honey production. There was an important study published considering this problem: Filipiak, M. (2019). Key pollen host plants provide balanced diets for wild bee larvae: A lesson for planting flower strips and hedgerows.
Journal of Applied Ecology, 56(6), 1410–1418. doi: 10.1111/1365-2664.13383 Lines 211-213 – does the cited publication analyze this issue and allows for such a categorical statement? What does “bees with special nesting behavior” mean? Solitary bees? Clarification and
details are needed. Conclusions and the Figure 2: I do not understand how the Authors came to the conclusions. This chapter again is disconnected from the rest of the text. It looks as the statement or an opinion, not like
conclusion, because the Authors not provide clear line of though leading to the conclusion. Considering the Figure 2, in my opinion this figure should appear in Introduction rather than in  Conclusion. It might introduce the Reader to the topics described further in the manuscript but it does
not provide any new knowledge. In concluding chapter I was expecting a reference to the title and explanation of how all the specific
examples, given before, relate to the bigger picture of wild bee conservation. In my opinion conclusion should be rethinked and rewritten to provide the reader: (1) the frame for easy understanding of what
was said before within the context of wild bee conservation in an anthropocentric world, (2) new knowledge gained via synergistic discussion of known phenomena (what we can learn based on
previous paragraphs and what new important knowledge is revealed; what is the Author’s take home message?) and (3) future directions – what knowledge is highly needed? What basic research should be done, which questions should we ask? How to apply what we already know for better future studies?

It is not easy to write valuable review paper and it is even harder to write novel review paper considering bees. I strongly encourage the Authors to consider recently published reviews on bees to gain knowledge on was what already said. Then the Authors may focus on specific aspects of bee
conservation that need clarification or are novel and promising and have not been sufficiently discussed in previous reviews. I think focusing on different national conservation strategies and providing information on ongoing research projects, as proposed above, may be valuable novel point
of view. The Authors should also focus on providing gripping and coherent story to the reader. To that end the manuscript needs well-planned outline and central topical concept. Providing the motivation for the review and explaining its contribution to the field is also needed. The core text should discuss the literature in synergistic and integrative manner to provide rather new point of view via synthesis of the literature, rather than just summary of the literature. Ultimately, the added value of the review
should be emphasized in conclusions.

Its my job as a reviewer to be critical and to show the way of improving the text, therefore I hope my comments will be considered as constructive. I congratulate the Authors for their work and encourage them to rewrite the text thoroughly to provide valuable and highly citable contribution to the field. I will be happy to cite this manuscript in my future studies if appropriately improved.

Author Response

The Authors have undertaken the difficult task of writing a review paper on the wild bee conservation. A lot has been already said regarding this topic and it is hard to provide novel and needed information to the Reader. Unfortunately, the Authors have taken the easy way and provided too much information that was already discussed in recent reviews, including in the review published in Insects by Belski and Neelendra (2019). The Authors should familiarize with this important paper to prevent replication of information already provided there.

-- We thoroughly read this review to (1) delete information that was already deeply treated in this paper but also to (2) complete our review with information that was lacking. You will find many details in the next comments/responses to illustrate these inputs.

Also, in the introductory part of the current review the Authors should briefly refer to recently published reviews and should clearly state what new and needed information they want to provide in the current review, why it is important and interesting to the Reader. For example the current review might focus on emerging issues and the newest literature. In the current version of the manuscript the Authors focused on well-established knowledge only scanning through some new research in chronicle record manner (i.e. not really discussing the meaning, importance and significance of recent advances) as well as describing marginally a few conservation initiatives, limited to the European Union, instead of providing real review of such initiatives.

-- We have now added many papers conducted during the last 5 years, e.g.:

  • Hall & Steiner, 2019, about pollinator conservation policy in USA (lines 228-231)
  • The meta-analysis of van Klink et al., 2020, about global decline of terrestrial insects (lines 50-51)
  • Cely-Santos & Philipott, 2019, about the structuring of wild bees by landscape in Colombia (lines 163-164)
  • Saunders et al., 2018, about the role of managed honeybees in bee conservation (lines 182-185)
  • Turo & Gardiner, 2019, about the potential of urban green spaces in wild bee conservation (lines 182-186)
  • Tonietto & Larkin, 2018, about the impact of habitat restoration on wild bee conservation (lines 200-202, 215-216)
  • Vaudo et al., 2018, about the impact of pollen nutritional intake on bumblebee colony growth (lines 269-271)
  • Lane et al., 2020, about the importance of floral resources in the restoration of prairies (lines 262-265)
  • Gérard et al., 2019, about the potential impact of fragmented habitats on bee body size (lines 252-254)
  • Geppert et al., 2020, about the impact of agri-environmental schemes on pollinators (lines 273-276)
  • Mallinger et al., 2019, about the optimization of plant mixtures for pollinators (lines 281-283)
  • Filipiak, 2018, about the importance of nutrional ecology for wild bee conservation (lines 303-305, 309-311, 539-540)
  • Gresty et al., 2018, about the importance of flower preference for cavity-nesting solitary bees (lines 306-307, 538-539)
  • Filipiak et al., 2017, about the importance to take into account ecological stoichiometry to conserve bees (lines 311-313)
  • Gradish et al., 2019, about the pesticide exposure of honey bees and bumble bees (lines 327-328)
  • Egan et al., 2020, about Integrated Pest and Pollinator Management (lines 342-346)
  • Sivakoff et al., 2018, about the impact of landscape urbanization on bee communities (lines 421-423)
  • Majewska & Alitzer, 2018, about the impact of plants in gardens on pollinators (lines 430-432)
  • Aizen et al., 2019, about the impact of bumblebee commercial trade on native species (lines 444-445)
  • Silva & Minor, 2017, about the relation between adolescents experience/knowledge and their attitudes towards bees (lines 486-488)
  • Schonfelder & Bogner, 2017, about individual perception of bees (lines 493-496)

We also mention more clearly in the aim of the review as well as in the conclusion the added-value of our study lines 72-78 as well as in more details in the conclusion (lines 519-562).

“More specifically, we highlighted the conservation measures that increase the quality of bee-friendly habitats among a wide range of landscapes, from semi-natural habitats to urban and agricultural landscapes. We underlined the needed floral and nesting resources, as well as possible habitat managements. We then tackled the case of alien species and ended up by the ways to communicate and educate the public. Throughout this review, we pointed out the recent projects and studies that tried to fill the numerous gaps of wild bee conservation.”

Information about projects conducted outside of European Union, as well as case-studies in USA or Brazil, for example, have been added as mentioned above, e.g. lines 163-165, 210-211, 263-265, 421-423, 441-447, 496-502, 547-547, …

In my opinion, the current version of the manuscript is not suitable for publication,
however it represents a good start - a really good draft on which the valuable paper may be build, after major rewriting of the text and elaborating some specific issues (as suggested below). Therefore, I suggest either resubmission of the improved version of the paper or major review, meaning deep and thorough rewriting of the paper, as suggested below. Since the manuscript, in my opinion, should be thoroughly rewritten, I provide only major comments.

I strongly encourage the Authors to familiarize with this short but helpful article before rewriting the manuscript: Sayer, E. J. (2018). The anatomy of an excellent review paper. Functional Ecology, 32(10), 2278– 2281. https://doi.org/10.1111/1365-2435.13207

-- We agree with the reviewer and have carefully considered the recommended paper in the global structure of our manuscript. We then focused on a “clear rationale and specific aim”, “gave an overview to set the context and presented information in a logical structure to support the central concepts and advance the field”. In our thematic, the main central concept to improve the conservation of biodiversity is about the identification of critical knowledge gaps, but also the needed conservation actions. As added value, we hope to motivate researchers to undertake future studies but also take to the next step.  

General major commentsThree main weak points of this paper, making it not a review but rather sort of opinion paper are: (1) very narrow focus on European Union (covering approx. half of Europe’s area and this half is very specific regarding landscapes, habitats, agriculture, law, politics, summing up to specific nature conservation issues) and French-speaking countries inside the Union, which makes the paper of local importance. The Authors should broaden their point of view in the improved version of the paper;

-- This is a good point, we decided to broaden the focus of our review including several projects conducted outside of French-speaking countries in Europe (e.g. lines 352-355) but also outside Europe (e.g. 176-178, and many other examples in the comment above). However, as stated in several papers, much of the conservation work in wild bees has been conducted in Europe and USA. This is indeed one of the major gaps to be fill: what is happening in Asia, South America, and in the corresponding habitats? We tried to underline the few information available in these areas, e.g. lines 163-164, 210-211, 441-447.

(2) the paper is very short as for review and neither of the topics covered by the Authors is sufficiently discussed (the majority of them is only mentioned without any discussion!). The Authors should focus on the most interesting and novel issues and discuss them deeply;

-- We tried to discussed more topics more deeply, the review is thus indeed much longer now.

More specifically, we elaborated much more our discussion about:

  • How new projects (e.g. like Poshbees) could help us to fill the gaps in wild bee conservation, e.g. lines 333-335, 352-366.
  • How we can chose floral resources, in flower seed mixtures for example, e.g. lines 271-283, 301-320
  • What are the obstacle to the understanding of a broader audience about the issues in wild bee conservation lines 493-502.
  • Potential impact of commercial bumblebee trade on native species lines 439-448.
  • The evaluation of agri-environmental schemes lines 271-283.
  • The addition of many studies and projects outside UE lines 163-165, 210-211, 263-265, 421-423, 441-447, 496-502, 547-547
  • Importance to conserve bees for their ecological role and their part in food webs lines 107-122.

(3) in the current form the manuscript is not a coherent and gripping story. It is rather a collection of stories where one story does not led to another. The Authors should provide the clear line of though leading to the clear, concrete and precise conclusions saying what we can learn based on previous paragraphs and what new important knowledge is revealed when discussing previous sections in holistic and synergistic approach. Content wise, to focus on new knowledge that this manuscript could provide, I have paid particular attention to broadening its substantive effect. First, I provide most important recent literature that seems to be ignored by the Authors. I recommend to familiarize with the below mentioned studies and reports:

Belsky, & Joshi. (2019). Impact of Biotic and Abiotic Stressors on Managed and Feral Bees. Insects, 10(8), 233. doi: 10.3390/insects10080233 – in-depth review regarding issues related to the bee conservation. It covers topics discussed in the current review, including a chapter on initiatives to support bees;

Ollerton, J. (2017). Pollinator Diversity: Distribution, Ecological Function, and Conservation. Annual Review of Ecology, Evolution, and Systematics, 48(1), 353–376. doi:10.1146/annurev-ecolsys110316-022919 – this paper was cited by the Authors but it seems that the Authors do not read it thoroughly. I strongly recommend detailed, close reading of this publication. The eco-evo context is necessary when discussing bee conservation (in the current text the Authors provide exclusively the economic/agricultural point of view and bees mean much more than this).

van Klink, R., Bowler, D. E., Gongalsky, K. B., Swengel, A. B., Gentile, A., & Chase, J. M. (2020). Meta-analysis reveals declines in terrestrial but increases in freshwater insect abundances. Science, 420(April), in press. doi: 10.1126/science.aax9931 – this paper should be considered when writing about insect decline.

-- We agree with this remark. We tried to write our paper following a common thread so that it can be read like a gripping story but it had to be improve. We have now tried to elaborate our manuscript as detailed below:

  1. On the assessment that there is a huge gap between knowledge and understanding of the public audience, and that we lack of empirical studies…
  2. Many measures to conserve bees have no positive effect, or we even do not know their real effects. Yet, more and more information, at different scales, showed us that overall, wild bees are declining.
  3. Based on this, we need a review that highlights the different measures, the ongoing projects on this topic as well as the gap and a future roadmap. We began to review why we should conserve them, as well as how do we know which species are under threat, on under which threat. Theses first parts are crucial to have a basis and to know the reasons to work on this topic.
  4. We then focused on the heart of our study: the measures and the actors of their conservation. We divided this in different sub-parts
  5. First, the conservation of the whole habitat, notably by creating protected areas and then, the importance of a wide variety of habitats and to take into account ecological traits of targeted species in the conservation measures
  6. A focus on urban habitats and agricultural habitats
  7. More specifically, the choice of floral resources in this wide range of habitats
  8. The use of chemicals and the potential alternatives
  9. Provide nesting resources
  10. The issue with alien species
  11. And finally, based on all these empirical evidence, how to communicate and educate people.
  12. We finally conclude on which new information brings our review, and the potential goals of future researches

These papers have also been (re)read and we included their important ideas that was lacking in our paper as mentioned above.

And here are some papers (and one report) considering ignored aspects of bee conservation (filling some gaps) and initiatives to support bees:
Reversing the Decline of Insects, ed: Dave Goulson, The Wildlife Trusts 2020: https://bit.ly/3iKuQdJ more: https://www.wildlifetrusts.org/take-action-insects Turo, K. J., & Gardiner, M. M. (2019). From potential to practical: conserving bees in urban public green spaces. Frontiers in Ecology and the Environment, 17(3), 167–175. doi: 10.1002/fee.2015

Wood, T. J., Holland, J. M., & Goulson, D. (2017). Providing foraging resources for solitary bees on farmland: current schemes for pollinators benefit a limited suite of species. Journal of Applied Ecology, 54(1), 323–333. doi:10.1111/1365-2664.12718

Egan, P. A., Dicks, L. V., Hokkanen, H. M. T., & Stenberg, J. A. (2020). Delivering Integrated Pest and Pollinator Management (IPPM). Trends in Plant Science, 25(6), 577–589. doi: 10.1016/j.tplants.2020.01.006

Gresty, C. E. A., Clare, E., Devey, D. S., Cowan, R. S., Csiba, L., Malakasi, P., … Willis, K. J. (2018). Flower preferences and pollen transport networks for cavity-nesting solitary bees: Implications for the design of agri-environment schemes. Ecology and Evolution, 8(15), 7574–7587. doi:10.1002/ece3.4234

Filipiak, M. (2018). A Better Understanding of Bee Nutritional Ecology Is Needed to Optimize Conservation Strategies for Wild Bees—The Application of Ecological Stoichiometry. Insects, 9(3), 85. doi:10.3390/insects9030085

Filipiak, M. (2019). Key pollen host plants provide balanced diets for wild bee larvae: A lesson for planting flower strips and hedgerows. Journal of Applied Ecology, 56(6), 1410–1418. doi: 10.1111/1365-2664.13383

Carvell, C., Bourke, A. F. G., Dreier, S., Freeman, S. N., Hulmes, S., Jordan, W. C., … Heard, M. S. (2017). Bumblebee family lineage survival is enhanced in high-quality landscapes. Nature, 543(7646), 547–549. doi:10.1038/nature21709

Hall, D. M., & Steiner, R. (2019). Insect pollinator conservation policy innovations at subnational levels: Lessons for lawmakers. Environmental Science & Policy, 93(October 2018), 118–128. doi:10.1016/j.envsci.2018.12.026

Majewska, A. A., & Altizer, S. (2020). Planting gardens to support insect pollinators. Conservation Biology, 34(1), 15–25. doi: 10.1111/cobi.13271

Schönfelder, M. L., & Bogner, F. X. (2017). Individual perception of bees: Between perceived danger and willingness to protect. PLOS ONE, 12(6), e0180168. doi:10.1371/journal.pone.0180168

Wilson, J. S., Forister, M. L., & Carril, O. M. (2017). Interest exceeds understanding in public support of bee conservation. Frontiers in Ecology and the Environment, 15(8), 460–466. doi:10.1002/fee.1531

Silva, A., & Minor, E. S. (2017). Adolescents’ Experience and Knowledge of, and Attitudes toward, Bees: Implications and Recommendations for Conservation. Anthrozoös, 30(1), 19–32. doi:10.1080/08927936.2017.1270587

Colla, S. R., & MacIvor, J. S. (2017). Questioning public perception, conservation policy, and recovery actions for honeybees in North America. Conservation Biology, 31(5), 1202–1204. doi:10.1111/cobi.12839

Tonietto, R. K., & Larkin, D. J. (2018). Habitat restoration benefits wild bees: A meta-analysis. Journal of Applied Ecology, 55(2), 582–590. doi:10.1111/1365-2664.13012

Lane, I. G., Herron‐Sweet, C. R., Portman, Z. M., & Cariveau, D. P. (2020). Floral resource diversity drives bee community diversity in prairie restorations along an agricultural landscape gradient. Journal of Applied Ecology, (January), 1365-2664.13694. doi: 10.1111/1365-2664.13694

Geppert, C., Hass, A., Földesi, R., Donkó, B., Akter, A., Tscharntke, T., & Batáry, P. (2020). Agrienvironment schemes enhance pollinator richness and abundance but bumblebee reproduction depends on field size. Journal of Applied Ecology, 1–11. doi: 10.1111/1365-2664.13682

-- Each of these papers have been carefully read and we tried to include the information that was lacking in our paper as mentioned in a previous answer.

Knowledge gaps are mentioned in the title. It may be interesting to discuss these gaps within the context of the ongoing research projects, which will provide new knowledge in the nearest future. The Authors could compare the projects and provide information on which knowledge gaps will be probably filled in the nearest future and on which gaps we should focus now when writing new research proposals. For example, I know that following projects are ongoing within the EU (it’s the Authors job to find more):

https://www.poshbee.eu/
https://nutrib2.project.uj.edu.pl/start
https://www.ecostack-h2020.eu/
https://ec.europa.eu/food/animals/live_animals/bees/research_en
and more, not strictly related to EU programs + worldwide:
http://worldbeeproject.org/
https://www.biodiversityireland.ie/
https://pollinators.ie/record-pollinators/bees/
https://ento.psu.edu/pollinators/stakeholder-advisory-board-on-pollinator-health

The Authors mention National Pollinator Strategy in the text. Since the manuscript is a review, it is not enough. The Authors should discuss various strategies related to pollinator conservation in different countries, compare them within the context of landscapes, habitats land use and agriculture characteristics, that differ between countries and ultimately the Authors should provide advice on the realities of pollinator conservation adoption.

-- This is a really good point. We focused much more on these projects to highlight the information that they could add in the future, to fill the gaps.

Lines 176-178, 182-185, 317-318, 333-335, 352-366.

Detailed major comments

Title is not informative: practically it means “wild bee conservation: a review”. I suggest to focus in the title on one strong take home message and make this message the central point of the manuscript.

 We totally agree with this comment and we tried to focus on the main take home message of this paper (and according to Sayer 2018). The title is now “Beyond the decline of wild bees: optimizing conservation measures and bringing together the actors”.

Introduction – the Authors write about “lack of awareness of the great diversity of pollinators” which may lead to “irrelevant actions”, misguided measures, etc. And the rest of the manuscript concerns only bees. Why? Why the Authors first write about the great diversity of pollinators and then ignore it and write only about bees?

-- This was simply to contextualize of study in the global pollinator crisis. This lack of awareness on pollinators reflects the lack of awareness on bees. We would like to keep this beginning of introduction on pollinators in their entirety, but if you found this really irrelevant, we will fix it in the next version.

Chapters 1 and 2 are disconnected from the rest of the text. The manuscript could start from the chapter 3. Figure 1 – description of abbreviations is needed. The chapter 2: “Why should we conserve wild bees?” should be either removed (it is not necessary) or elaborated much more and completely rewritten. The Authors described only the commercial/agricultural importance of wild bees, which is very anthropocentric. Bees are much more than this – they play ecological roles in their environments and interact with different components of food webs, etc.

-- The description of the abbreviations have been added.

We also elaborated much more the chapter 2. We think that, even if it has already been treated, a minimum information about the importance of bees has to be included in our paper because it is why we have to lead conservation measures. We thus tried to find the balance and include information on their ecological roles and their crucial importance in ecological networks / food web. Lines 107-122.

The topic of wild bee conservation initiatives is central part of the title but was neglected in the text of the manuscript: the LIFE EU framework was mentioned in lines 135-149 and all the information provided in lines 150-205 are mainly hand-waiving, i.e. superficial, round-about and without any specifics. This should be improved.

-- This is true that we mostly focused on the global conservation measures (supply for food resources, nesting resources) more than the conservation initiatives at the national/continental levels. Thanks to the projects you advised and our additional overview of the literature, we have been able to improve this part as described in the comments above.

Lines 170-195 and 333-337: the problem of appropriate floral composition of flower strips, beefriendly seed mixes etc. is very important. It seems that the majority of seed/flower mixes used are of not appropriate composition for wild bees, focusing on nectar-rich plants for honey production. There was an important study published considering this problem: Filipiak, M. (2019). Key pollen host plants provide balanced diets for wild bee larvae: A lesson for planting flower strips and hedgerows. Journal of Applied Ecology, 56(6), 1410–1418. doi: 10.1111/1365-2664.13383

-- The problem of appropriate floral composition of flower strips is now more widely treated, including the importance of ecological Stoichiometry developed by Filipiak lines 303-316.

Lines 211-213 – does the cited publication analyze this issue and allows for such a categorical statement? What does “bees with special nesting behavior” mean? Solitary bees? Clarification and details are needed.

-- The cited publication is the Belgium Red List of Bees and then assess the probability of extinction of each bee species following the IUCN methodology. Considering the nesting requirements, it has been observed that some bee groups were more vulnerable than others. We clarified and detailed the targeted sentence. Lines 374-375

Conclusions and the Figure 2: I do not understand how the Authors came to the conclusions. This chapter again is disconnected from the rest of the text. It looks as the statement or an opinion, not like conclusion, because the Authors not provide clear line of though leading to the conclusion.

Considering the Figure 2, in my opinion this figure should appear in Introduction rather than in Conclusion. It might introduce the Reader to the topics described further in the manuscript but it does not provide any new knowledge.

-- Here you will find the answer concerning figure 2. In the comment below, the answer concerning the conclusion.

We agree, the figure 2 is now in the end of the introduction and introduce the topics that we discussed below.

In concluding chapter I was expecting a reference to the title and explanation of how all the specific examples, given before, relate to the bigger picture of wild bee conservation. In my opinion conclusion should be rethinked and rewritten to provide the reader: (1) the frame for easy understanding of what was said before within the context of wild bee conservation in an anthropocentric world, (2) new knowledge gained via synergistic discussion of known phenomena (what we can learn based on previous paragraphs and what new important knowledge is revealed; what is the Author’s take home message?) and (3) future directions – what knowledge is highly needed? What basic research should be done, which questions should we ask? How to apply what we already know for better future studies?

-- Thanks to your comment, the conclusion has been entirely reshaped lines 519-571. We underlined each novelties and take home messages that we think should act as a roadmap for future works in wild bee conservation. Specifically, we underlined that one central point that is lacking in most conservation actions, is that targeted species are not clearly identified and that results on honeybees and bumblebees are often used as proxy. However, specific responses depend on a bunch of ecological traits; many requirements of wild bee species are totally different from those of bumblebees and honeybees. It is perfectly illustrated by the choice of flower in seed mixes. We also addressed the fact that, in a more and more disturbed world, some alternatives like exotic species (punctually and depending on the context) could help some population to recover. We highlighted the huge disequilibrium between the studies about floral resources and nesting resources – an additional basic researches could be done on this point. This is also true for conservation methods outside of Europe and USA, as well as in more specific ecosystems like arctic regions. Finally, in parallel to these researches, it is crucial to integrate work packages in our research projects, focusing on the educational and communication outcomes of our results.

It is not easy to write valuable review paper and it is even harder to write novel review paper considering bees. I strongly encourage the Authors to consider recently published reviews on bees to gain knowledge on was what already said. Then the Authors may focus on specific aspects of bee conservation that need clarification or are novel and promising and have not been sufficiently discussed in previous reviews. I think focusing on different national conservation strategies and providing information on ongoing research projects, as proposed above, may be valuable novel point of view. The Authors should also focus on providing gripping and coherent story to the reader. To that end the manuscript needs well-planned outline and central topical concept. Providing the motivation for the review and explaining its contribution to the field is also needed. The core text should discuss the literature in synergistic and integrative manner to provide rather new point of view via synthesis of the literature, rather than just summary of the literature. Ultimately, the added value of the review should be emphasized in conclusions.

-- We really appreciate the huge work you produced (and probably the huge amount of time) to improve our review, and the valuable information that you added. We hope that this new version will fit with your expectations.

Reviewer 3 Report

The review presented is summarizing our knowledge on initiatives to halt bee decline and knowledge gaps of wild bee decline and biology.

Though the idea is not exceptional, as there are other reviews dealing with this topic it does give a summary of our current knowledge and knowledge gaps of  wild bee status in general and the conservation methods and initiative used so far in Europe. 

The conclusions are in line with the current views of the scientific community, namely: better assessment of the status of various less popular bee species, research on the drivers of decline especially in less studied groups and proposing strategies and tools for conservation. 

There is one major weakness of this review: if it is to be "a catalyst to implement concrete and qualitative conversation actions for bees" than it should be more global and less EU centered. In this form it could be a catalyst for EU actions and research, but not a global review.

I would either widen the perspective of the review to include also actions and examples from other continents, including also some indigenous actions and initiatives or stick to the EU and make it an EU review and catalyst. In the second case, also some additional EU examples would be good, currently it is too France and Belgium centered.

Author Response

The review presented is summarizing our knowledge on initiatives to halt bee decline and knowledge gaps of wild bee decline and biology. Though the idea is not exceptional, as there are other reviews dealing with this topic it does give a summary of our current knowledge and knowledge gaps of wild bee status in general and the conservation methods and initiative used so far in Europe. The conclusions are in line with the current views of the scientific community, namely: better assessment of the status of various less popular bee species, research on the drivers of decline especially in less studied groups and proposing strategies and tools for conservation. 

-- Many thanks for this comment

There is one major weakness of this review: if it is to be "a catalyst to implement concrete and qualitative conversation actions for bees" than it should be more global and less EU centered. In this form it could be a catalyst for EU actions and research, but not a global review. I would either widen the perspective of the review to include also actions and examples from other continents, including also some indigenous actions and initiatives or stick to the EU and make it an EU review and catalyst. In the second case, also some additional EU examples would be good, currently it is too France and Belgium centered.

-- You are completely right. We thoroughly review a new batch of studies in the literature about what is done elsewhere in the world, including different projects notably in America. Among them, here are some examples of what we added: e.g. lines 163-165, 210-211, 263-265, 421-423, 441-447, 496-502, 547-547 ,…
